# Convolutional Kernel Networks

**Julien Mairal, Piotr Koniusz, Zaid Harchaoui, and Cordelia Schmid**
Inria*
firstname.lastname@inria.fr

## Abstract

An important goal in visual recognition is to devise image representations that are invariant to particular transformations. In this paper, we address this goal with a new type of convolutional neural network (CNN) whose invariance is encoded by a reproducing kernel. Unlike traditional approaches where neural networks are learned either to represent data or for solving a classification task, our network learns to approximate the kernel feature map on training data.

Such an approach enjoys several benefits over classical ones. First, by teaching CNNs to be invariant, we obtain simple network architectures that achieve a similar accuracy to more complex ones, while being easy to train and robust to overfitting. Second, we bridge a gap between the neural network literature and kernels, which are natural tools to model invariance. We evaluate our methodology on visual recognition tasks where CNNs have proven to perform well, *e.g.*, digit recognition with the MNIST dataset, and the more challenging CIFAR-10 and STL-10 datasets, where our accuracy is competitive with the state of the art.

## 1 Introduction

We have recently seen a revival of attention given to convolutional neural networks (CNNs) [22] due to their high performance for large-scale visual recognition tasks [15, 21, 30]. The architecture of CNNs is relatively simple and consists of successive layers organized in a hierarchical fashion; each layer involves convolutions with learned filters followed by a pointwise non-linearity and a downsampling operation called "feature pooling". The resulting image representation has been empirically observed to be invariant to image perturbations and to encode complex visual patterns [33], which are useful properties for visual recognition. Training CNNs remains however difficult since high-capacity networks may involve billions of parameters to learn, which requires both high computational power, *e.g.*, GPUs, and appropriate regularization techniques [18, 21, 30].

The exact nature of invariance that CNNs exhibit is also not precisely understood. Only recently, the invariance of related architectures has been characterized; this is the case for the wavelet scattering transform [8] or the hierarchical models of [7]. Our work revisits convolutional neural networks, but we adopt a significantly different approach than the traditional one. Indeed, we use kernels [26], which are natural tools to model invariance [14]. Inspired by the hierarchical kernel descriptors of [2], we propose a reproducing kernel that produces multi-layer image representations.

Our main contribution is an approximation scheme called *convolutional kernel network* (CKN) to make the kernel approach computationally feasible. Our approach is a new type of unsupervised convolutional neural network that is trained to approximate the kernel map. Interestingly, our network uses non-linear functions that resemble rectified linear units [1, 30], even though they were not handcrafted and naturally emerge from an approximation scheme of the Gaussian kernel map.

By bridging a gap between kernel methods and neural networks, we believe that we are opening a fruitful research direction for the future. Our network is learned without supervision since the

label information is only used subsequently in a support vector machine (SVM). Yet, we achieve competitive results on several datasets such as MNIST [22], CIFAR-10 [20] and STL-10 [13] with simple architectures, few parameters to learn, and no data augmentation. Open-source code for learning our convolutional kernel networks is available on the first author's webpage.

## 1.1 Related Work

There have been several attempts to build kernel-based methods that mimic deep neural networks; we only review here the ones that are most related to our approach.

**Arc-cosine kernels.** Kernels for building deep large-margin classifiers have been introduced in [10]. The multilayer arc-cosine kernel is built by successive kernel compositions, and each layer relies on an integral representation. Similarly, our kernels rely on an integral representation, and enjoy a multilayer construction. However, in contrast to arc-cosine kernels: (i) we build our sequence of kernels by *convolutions*, using local information over spatial neighborhoods (as opposed to compositions, using global information); (ii) we propose a new training procedure for learning a compact representation of the kernel in a *data-dependent* manner.

**Multilayer derived kernels.** Kernels with invariance properties for visual recognition have been proposed in [7]. Such kernels are built with a parameterized "neural response" function, which consists in computing the maximal response of a base kernel over a local neighborhood. Multiple layers are then built by iteratively renormalizing the response kernels and pooling using neural response functions. Learning is performed by plugging the obtained kernel in an SVM. In contrast to [7], we propagate information up, from lower to upper layers, by using sequences of convolutions. Furthermore, we propose a simple and effective data-dependent way to learn a compact representation of our kernels and show that we obtain near state-of-the-art performance on several benchmarks.

**Hierarchical kernel descriptors.** The kernels proposed in [2, 3] produce multilayer image representations for visual recognition tasks. We discuss in details these kernels in the next section: our paper generalizes them and establishes a strong link with convolutional neural networks.

## 2 Convolutional Multilayer Kernels

The convolutional multilayer kernel is a generalization of the hierarchical kernel descriptors introduced in computer vision [2, 3]. The kernel produces a sequence of image representations that are built on top of each other in a multilayer fashion. Each layer can be interpreted as a non-linear transformation of the previous one with additional spatial invariance. We call these layers *image feature maps*[1], and formally define them as follows:

**Definition 1.** *An image feature map $\varphi$ is a function $\varphi : \Omega \to \mathcal{H}$, where $\Omega$ is a (usually discrete) subset of $[0,1]^d$ representing normalized "coordinates" in the image and $\mathcal{H}$ is a Hilbert space.*

For all practical examples in this paper, $\Omega$ is a two-dimensional grid and corresponds to different locations in a two-dimensional image. In other words, $\Omega$ is a set of pixel coordinates. Given $\mathbf{z}$ in $\Omega$, the point $\varphi(\mathbf{z})$ represents some characteristics of the image at location $\mathbf{z}$, or in a neighborhood of $\mathbf{z}$. For instance, a color image of size $m \times n$ with three channels, red, green, and blue, may be represented by an initial feature map $\varphi_0 : \Omega_0 \to \mathcal{H}_0$, where $\Omega_0$ is an $m \times n$ regular grid, $\mathcal{H}_0$ is the Euclidean space $\mathbb{R}^3$, and $\varphi_0$ provides the color pixel values. With the multilayer scheme, non-trivial feature maps will be obtained subsequently, which will encode more complex image characteristics. With this terminology in hand, we now introduce the convolutional kernel, first, for a single layer.

**Definition 2** (**Convolutional Kernel with Single Layer**). *Let us consider two images represented by two image feature maps, respectively $\varphi$ and $\varphi' : \Omega \to \mathcal{H}$, where $\Omega$ is a set of pixel locations, and $\mathcal{H}$ is a Hilbert space. The one-layer convolutional kernel between $\varphi$ and $\varphi'$ is defined as*

$$K(\varphi, \varphi') := \sum_{\mathbf{z} \in \Omega} \sum_{\mathbf{z}' \in \Omega} \|\varphi(\mathbf{z})\|_{\mathcal{H}} \|\varphi'(\mathbf{z}')\|_{\mathcal{H}} \, e^{-\frac{1}{2\beta^2} \|\mathbf{z} - \mathbf{z}'\|_2^2} e^{-\frac{1}{2\sigma^2} \|\tilde{\varphi}(\mathbf{z}) - \tilde{\varphi}'(\mathbf{z}')\|_{\mathcal{H}}^2}, \quad (1)$$

*where $\beta$ and $\sigma$ are smoothing parameters of Gaussian kernels, and $\tilde{\varphi}(\mathbf{z}) := (1/\|\varphi(\mathbf{z})\|_{\mathcal{H}})\,\varphi(\mathbf{z})$ if $\varphi(\mathbf{z}) \neq 0$ and $\tilde{\varphi}(\mathbf{z}) = 0$ otherwise. Similarly, $\tilde{\varphi}'(\mathbf{z}')$ is a normalized version of $\varphi'(\mathbf{z}')$.[2]*

It is easy to show that the kernel $K$ is positive definite (see Appendix A). It consists of a sum of pairwise comparisons between the image features $\varphi(\mathbf{z})$ and $\varphi'(\mathbf{z}')$ computed at all spatial locations $\mathbf{z}$ and $\mathbf{z}'$ in $\Omega$. To be significant in the sum, a comparison needs the corresponding $\mathbf{z}$ and $\mathbf{z}'$ to be close in $\Omega$, and the normalized features $\tilde{\varphi}(\mathbf{z})$ and $\tilde{\varphi}'(\mathbf{z}')$ to be close in the feature space $\mathcal{H}$. The parameters $\beta$ and $\sigma$ respectively control these two definitions of "closeness". Indeed, when $\beta$ is large, the kernel $K$ is invariant to the positions $\mathbf{z}$ and $\mathbf{z}'$ but when $\beta$ is small, only features placed at the same location $\mathbf{z} = \mathbf{z}'$ are compared to each other. Therefore, the role of $\beta$ is to control how much the kernel is locally shift-invariant. Next, we will show how to go beyond one single layer, but before that, we present concrete examples of simple input feature maps $\varphi_0 : \Omega_0 \to \mathcal{H}_0$.

**Gradient map.** Assume that $\mathcal{H}_0 = \mathbb{R}^2$ and that $\varphi_0(\mathbf{z})$ provides the two-dimensional gradient of the image at pixel $\mathbf{z}$, which is often computed with first-order differences along each dimension. Then, the quantity $\|\varphi_0(\mathbf{z})\|_{\mathcal{H}_0}$ is the gradient intensity, and $\tilde{\varphi}_0(\mathbf{z})$ is its orientation, which can be characterized by a particular angle—that is, there exists $\theta$ in $[0; 2\pi]$ such that $\tilde{\varphi}_0(\mathbf{z}) = [\cos(\theta), \sin(\theta)]$. The resulting kernel $K$ is exactly the kernel descriptor introduced in [2, 3] for natural image patches.

**Patch map.** In that setting, $\varphi_0$ associates to a location $\mathbf{z}$ an image patch of size $m \times m$ centered at $\mathbf{z}$. Then, the space $\mathcal{H}_0$ is simply $\mathbb{R}^{m \times m}$, and $\tilde{\varphi}_0(\mathbf{z})$ is a *contrast-normalized* version of the patch, which is a useful transformation for visual recognition according to classical findings in computer vision [19]. When the image is encoded with three color channels, patches are of size $m \times m \times 3$.

We now define the multilayer convolutional kernel, generalizing some ideas of [2].

**Definition 3 (Multilayer Convolutional Kernel).** *Let us consider a set $\Omega_{k-1} \subseteq [0,1]^d$ and a Hilbert space $\mathcal{H}_{k-1}$. We build a new set $\Omega_k$ and a new Hilbert space $\mathcal{H}_k$ as follows:*

*(i) choose a patch shape $\mathcal{P}_k$ defined as a bounded symmetric subset of $[-1,1]^d$, and a set of coordinates $\Omega_k$ such that for all location $\mathbf{z}_k$ in $\Omega_k$, the patch $\{\mathbf{z}_k\} + \mathcal{P}_k$ is a subset of $\Omega_{k-1}$;[3] In other words, each coordinate $\mathbf{z}_k$ in $\Omega_k$ corresponds to a valid patch in $\Omega_{k-1}$ centered at $\mathbf{z}_k$.*

*(ii) define the convolutional kernel $K_k$ on the "patch" feature maps $\mathcal{P}_k \to \mathcal{H}_{k-1}$, by replacing in (1): $\Omega$ by $\mathcal{P}_k$, $\mathcal{H}$ by $\mathcal{H}_{k-1}$, and $\sigma, \beta$ by appropriate smoothing parameters $\sigma_k, \beta_k$. We denote by $\mathcal{H}_k$ the Hilbert space for which the positive definite kernel $K_k$ is reproducing.*

*An image represented by a feature map $\varphi_{k-1} : \Omega_{k-1} \to \mathcal{H}_{k-1}$ at layer $k$–1 is now encoded in the $k$-th layer as $\varphi_k : \Omega_k \to \mathcal{H}_k$, where for all $\mathbf{z}_k$ in $\Omega_k$, $\varphi_k(\mathbf{z}_k)$ is the representation in $\mathcal{H}_k$ of the patch feature map $\mathbf{z} \mapsto \varphi_{k-1}(\mathbf{z}_k + \mathbf{z})$ for $\mathbf{z}$ in $\mathcal{P}_k$.*

Concretely, the kernel $K_k$ between two patches of $\varphi_{k-1}$ and $\varphi'_{k-1}$ at respective locations $\mathbf{z}_k$ and $\mathbf{z}'_k$ is

$$\sum_{\mathbf{z} \in \mathcal{P}_k} \sum_{\mathbf{z}' \in \mathcal{P}_k} \|\varphi_{k-1}(\mathbf{z}_k + \mathbf{z})\| \, \|\varphi'_{k-1}(\mathbf{z}'_k + \mathbf{z}')\| \, e^{-\frac{1}{2\beta_k^2}\|\mathbf{z}-\mathbf{z}'\|_2^2} e^{-\frac{1}{2\sigma_k^2}\|\tilde{\varphi}_{k-1}(\mathbf{z}_k+\mathbf{z})-\tilde{\varphi}'_{k-1}(\mathbf{z}'_k+\mathbf{z}')\|^2}, \quad (2)$$

where $\|.\|$ is the Hilbertian norm of $\mathcal{H}_{k-1}$. In Figure 1(a), we illustrate the interactions between the sets of coordinates $\Omega_k$, patches $\mathcal{P}_k$, and feature spaces $\mathcal{H}_k$ across layers. For two-dimensional grids, a typical patch shape is a square, for example $\mathcal{P} := \{-1/n, 0, 1/n\} \times \{-1/n, 0, 1/n\}$ for a $3 \times 3$ patch in an image of size $n \times n$. Information encoded in the $k$-th layer differs from the $(k-1)$-th one in two aspects: first, each point $\varphi_k(\mathbf{z}_k)$ in layer $k$ contains information about several points from the $(k-1)$-th layer and can possibly represent larger patterns; second, the new feature map is more locally shift-invariant than the previous one due to the term involving the parameter $\beta_k$ in (2).

The multilayer convolutional kernel slightly differs from the hierarchical kernel descriptors of [2] but exploits similar ideas. Bo et al. [2] define indeed several ad hoc kernels for representing local information in images, such as gradient, color, or shape. These kernels are close to the one defined in (1) but with a few variations. Some of them do not use normalized features $\tilde{\varphi}(\mathbf{z})$, and these kernels use different weighting strategies for the summands of (1) that are specialized to the image modality, *e.g.*, color, or gradient, whereas we use the same weight $\|\varphi(\mathbf{z})\|_{\mathcal{H}} \|\varphi'(\mathbf{z}')\|_{\mathcal{H}}$ for all kernels. The generic formulation (1) that we propose may be useful per se, but our main contribution comes in the next section, where we use the kernel as a new tool for learning convolutional neural networks.

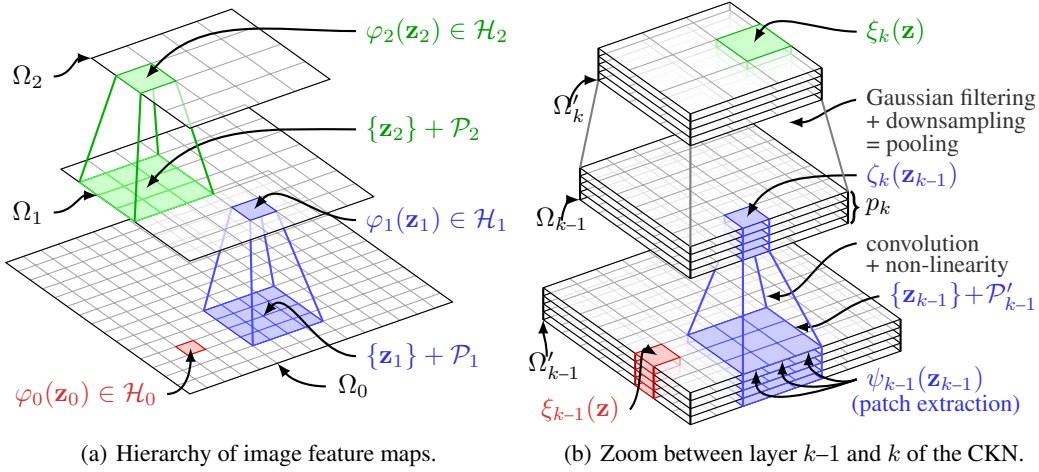

(a) Hierarchy of image feature maps.  (b) Zoom between layer $k–1$ and $k$ of the CKN.

Figure 1: Left: concrete representation of the successive layers for the multilayer convolutional kernel. Right: one layer of the convolutional neural network that approximates the kernel.

## 3  Training Invariant Convolutional Kernel Networks

Generic schemes have been proposed for approximating a non-linear kernel with a linear one, such as the Nyström method and its variants [5, 31], or random sampling techniques in the Fourier domain for shift-invariant kernels [24]. In the context of convolutional multilayer kernels, such an approximation is critical because computing the full kernel matrix on a database of images is computationally infeasible, even for a moderate number of images ($\approx 10\,000$) and moderate number of layers. For this reason, Bo et al. [2] use the Nyström method for their hierarchical kernel descriptors.

In this section, we show that when the coordinate sets $\Omega_k$ are two-dimensional regular grids, a natural approximation for the multilayer convolutional kernel consists of a sequence of spatial convolutions with learned filters, pointwise non-linearities, and pooling operations, as illustrated in Figure 1(b). More precisely, our scheme approximates the kernel map of $K$ defined in (1) at layer $k$ by finite-dimensional spatial maps $\xi_k : \Omega'_k \to \mathbb{R}^{p_k}$, where $\Omega'_k$ is a set of coordinates related to $\Omega_k$, and $p_k$ is a positive integer controlling the quality of the approximation. Consider indeed two images represented at layer $k$ by image feature maps $\varphi_k$ and $\varphi'_k$, respectively. Then,

**(A)** the corresponding maps $\xi_k$ and $\xi'_k$ are learned such that $K(\varphi_{k-1}, \varphi'_{k-1}) \approx \langle \xi_k, \xi'_k \rangle$, where $\langle ., . \rangle$ is the Euclidean inner-product acting as if $\xi_k$ and $\xi'_k$ were vectors in $\mathbb{R}^{|\Omega'_k| p_k}$;

**(B)** the set $\Omega'_k$ is linked to $\Omega_k$ by the relation $\Omega'_k = \Omega_k + \mathcal{P}'_k$ where $\mathcal{P}'_k$ is a patch shape, and the quantities $\varphi_k(\mathbf{z}_k)$ in $\mathcal{H}_k$ admit finite-dimensional approximations $\psi_k(\mathbf{z}_k)$ in $\mathbb{R}^{|\mathcal{P}'_k| p_k}$; as illustrated in Figure 1(b), $\psi_k(\mathbf{z}_k)$ is a patch from $\xi_k$ centered at location $\mathbf{z}_k$ with shape $\mathcal{P}'_k$;

**(C)** an activation map $\zeta_k : \Omega_{k-1} \mapsto \mathbb{R}^{p_k}$ is computed from $\xi_{k-1}$ by convolution with $p_k$ filters followed by a non-linearity. The subsequent map $\xi_k$ is obtained from $\zeta_k$ by a pooling operation.

We call this approximation scheme a convolutional kernel network (CKN). In comparison to CNNs, our approach enjoys similar benefits such as efficient prediction at test time, and involves the same set of hyper-parameters: number of layers, numbers of filters $p_k$ at layer $k$, shape $\mathcal{P}'_k$ of the filters, sizes of the feature maps. The other parameters $\beta_k, \sigma_k$ can be automatically chosen, as discussed later. Training a CKN can be argued to be as simple as training a CNN in an unsupervised manner [25] since we will show that the main difference is in the cost function that is optimized.

### 3.1  Fast Approximation of the Gaussian Kernel

A key component of our formulation is the Gaussian kernel. We start by approximating it by a linear operation with learned filters followed by a pointwise non-linearity. Our starting point is the next lemma, which can be obtained after a simple calculation.

**Lemma 1** (**Linear expansion of the Gaussian Kernel**). *For all $\mathbf{x}$ and $\mathbf{x}'$ in $\mathbb{R}^m$, and $\sigma > 0$,*

$$e^{-\frac{1}{2\sigma^2}\|\mathbf{x}-\mathbf{x}'\|_2^2} = \left(\frac{2}{\pi\sigma^2}\right)^{\frac{m}{2}} \int_{\mathbf{w}\in\mathbb{R}^m} e^{-\frac{1}{\sigma^2}\|\mathbf{x}-\mathbf{w}\|_2^2} e^{-\frac{1}{\sigma^2}\|\mathbf{x}'-\mathbf{w}\|_2^2} d\mathbf{w}. \tag{3}$$

The lemma gives us a mapping of any $\mathbf{x}$ in $\mathbb{R}^m$ to the function $\mathbf{w} \mapsto \sqrt{C}e^{-(1/\sigma^2)\|\mathbf{x}-\mathbf{w}\|_2^2}$ in $L^2(\mathbb{R}^m)$, where the kernel is linear, and $C$ is the constant in front of the integral. To obtain a finite-dimensional representation, we need to approximate the integral with a weighted finite sum, which is a classical problem arising in statistics (see [29] and chapter 8 of [6]). Then, we consider two different cases.

**Small dimension, $m \leq 2$.** When the data lives in a compact set of $\mathbb{R}^m$, the integral in (3) can be approximated by uniform sampling over a large enough set. We choose such a strategy for two types of kernels from Eq. (1): (i) the spatial kernels $e^{-\left(\frac{1}{2\beta^2}\right)\|\mathbf{z}-\mathbf{z}'\|_2^2}$; (ii) the terms $e^{-\left(\frac{1}{2\sigma^2}\right)\|\tilde{\varphi}(\mathbf{z})-\tilde{\varphi}'(\mathbf{z}')\|_{\mathcal{H}}^2}$ when $\varphi$ is the "gradient map" presented in Section 2. In the latter case, $\mathcal{H} = \mathbb{R}^2$ and $\tilde{\varphi}(\mathbf{z})$ is the gradient orientation. We typically sample a few orientations as explained in Section 4.

**Higher dimensions.** To prevent the curse of dimensionality, we learn to approximate the kernel on training data, which is intrinsically low-dimensional. We optimize importance weights $\boldsymbol{\eta} = [\eta_l]_{l=1}^p$ in $\mathbb{R}_+^p$ and sampling points $\mathbf{W} = [\mathbf{w}_l]_{l=1}^p$ in $\mathbb{R}^{m\times p}$ on $n$ training pairs $(\mathbf{x}_i, \mathbf{y}_i)_{i=1,\ldots,n}$ in $\mathbb{R}^m \times \mathbb{R}^m$:

$$\min_{\boldsymbol{\eta}\in\mathbb{R}_+^p, \mathbf{W}\in\mathbb{R}^{m\times p}} \left[\frac{1}{n}\sum_{i=1}^n \left(e^{-\frac{1}{2\sigma^2}\|\mathbf{x}_i-\mathbf{y}_i\|_2^2} - \sum_{l=1}^p \eta_l e^{-\frac{1}{\sigma^2}\|\mathbf{x}_i-\mathbf{w}_l\|_2^2} e^{-\frac{1}{\sigma^2}\|\mathbf{y}_i-\mathbf{w}_l\|_2^2}\right)^2\right]. \tag{4}$$

Interestingly, we may already draw some links with neural networks. When applied to unit-norm vectors $\mathbf{x}_i$ and $\mathbf{y}_i$, problem (4) produces sampling points $\mathbf{w}_l$ whose norm is close to one. After learning, a new unit-norm point $\mathbf{x}$ in $\mathbb{R}^m$ is mapped to the vector $[\sqrt{\eta_l}e^{-(1/\sigma^2)\|\mathbf{x}-\mathbf{w}_l\|_2^2}]_{l=1}^p$ in $\mathbb{R}^p$, which may be written as $[f(\mathbf{w}_l^\top\mathbf{x})]_{l=1}^p$, assuming that the norm of $\mathbf{w}_l$ is always one, where $f$ is the function $u \mapsto e^{(2/\sigma^2)(u-1)}$ for $u = \mathbf{w}_l^\top\mathbf{x}$ in $[-1,1]$. Therefore, the finite-dimensional representation of $\mathbf{x}$ only involves a linear operation followed by a non-linearity, as in typical neural networks. In Figure 2, we show that the shape of $f$ resembles the "rectified linear unit" function [30].

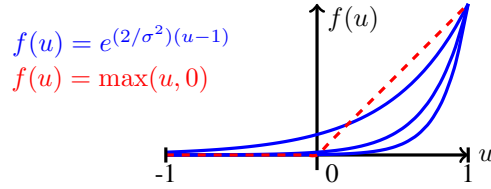

$$f(u) = e^{(2/\sigma^2)(u-1)}$$
$$f(u) = \max(u, 0)$$

Figure 2: In dotted red, we plot the "rectified linear unit" function $u \mapsto \max(u, 0)$. In blue, we plot non-linear functions of our network for typical values of $\sigma$ that we use in our experiments.

### 3.2 Approximating the Multilayer Convolutional Kernel

We have now all the tools in hand to build our convolutional kernel network. We start by making assumptions on the input data, and then present the learning scheme and its approximation principles.

**The zeroth layer.** We assume that the input data is a finite-dimensional map $\xi_0 : \Omega_0' \to \mathbb{R}^{p_0}$, and that $\varphi_0 : \Omega_0 \to \mathcal{H}_0$ "extracts" patches from $\xi_0$. Formally, there exists a patch shape $\mathcal{P}_0'$ such that $\Omega_0' = \Omega_0 + \mathcal{P}_0'$, $\mathcal{H}_0 = \mathbb{R}^{p_0|\mathcal{P}_0'|}$, and for all $\mathbf{z}_0$ in $\Omega_0$, $\varphi_0(\mathbf{z}_0)$ is a patch of $\xi_0$ centered at $\mathbf{z}_0$. Then, property **(B)** described at the beginning of Section 3 is satisfied for $k = 0$ by choosing $\psi_0 = \varphi_0$. The examples of input feature maps given earlier satisfy this finite-dimensional assumption: for the gradient map, $\xi_0$ is the gradient of the image along each direction, with $p_0 = 2$, $\mathcal{P}_0' = \{0\}$ is a $1\times 1$ patch, $\Omega_0 = \Omega_0'$, and $\varphi_0 = \xi_0$; for the patch map, $\xi_0$ is the input image, say with $p_0 = 3$ for RGB data.

**The convolutional kernel network.** The zeroth layer being characterized, we present in Algorithms 1 and 2 the subsequent layers and how to learn their parameters in a feedforward manner. It is interesting to note that the input parameters of the algorithm are exactly the same as a CNN—that is, number of layers and filters, sizes of the patches and feature maps (obtained here via the subsampling factor). Ultimately, CNNs and CKNs only differ in the cost function that is optimized for learning the filters and in the choice of non-linearities. As we show next, there exists a link between the parameters of a CKN and those of a convolutional multilayer kernel.

---

**Algorithm 1** Convolutional kernel network - learning the parameters of the $k$-th layer.

---

**input** $\xi_{k-1}^1, \xi_{k-1}^2, \ldots : \Omega_{k-1}' \to \mathbb{R}^{p_{k-1}}$ (sequence of $(k-1)$-th maps obtained from training images); $\mathcal{P}_{k-1}'$ (patch shape); $p_k$ (number of filters); $n$ (number of training pairs);

1: extract at random $n$ pairs $(\mathbf{x}_i, \mathbf{y}_i)$ of patches with shape $\mathcal{P}_{k-1}'$ from the maps $\xi_{k-1}^1, \xi_{k-1}^2, \ldots$;

2: if not provided by the user, set $\sigma_k$ to the 0.1 quantile of the data $(\|\mathbf{x}_i - \mathbf{y}_i\|_2)_{i=1}^n$;

3: **unsupervised learning:** optimize (4) to obtain the filters $\mathbf{W}_k$ in $\mathbb{R}^{|\mathcal{P}_{k-1}'|p_{k-1} \times p_k}$ and $\boldsymbol{\eta}_k$ in $\mathbb{R}^{p_k}$;

**output** $\mathbf{W}_k$, $\boldsymbol{\eta}_k$, and $\sigma_k$ (smoothing parameter);

---

**Algorithm 2** Convolutional kernel network - computing the $k$-th map form the $(k-1)$-th one.

---

**input** $\xi_{k-1} : \Omega_{k-1}' \to \mathbb{R}^{p_{k-1}}$ (input map); $\mathcal{P}_{k-1}'$ (patch shape); $\gamma_k \geq 1$ (subsampling factor); $p_k$ (number of filters); $\sigma_k$ (smoothing parameter); $\mathbf{W}_k = [\mathbf{w}_{kl}]_{l=1}^{p_k}$ and $\boldsymbol{\eta}_k = [\eta_{kl}]_{l=1}^{p_k}$ (layer parameters);

1: **convolution and non-linearity:** define the activation map $\zeta_k : \Omega_{k-1} \to \mathbb{R}^{p_k}$ as

$$\zeta_k : \mathbf{z} \mapsto \|\psi_{k-1}(\mathbf{z})\|_2 \left[ \sqrt{\eta_{kl}} e^{-\frac{1}{\sigma_k^2}\|\tilde{\psi}_{k-1}(\mathbf{z})-\mathbf{w}_{kl}\|_2^2} \right]_{l=1}^{p_k}, \tag{5}$$

where $\psi_{k-1}(\mathbf{z})$ is a vector representing a patch from $\xi_{k-1}$ centered at $\mathbf{z}$ with shape $\mathcal{P}_{k-1}'$, and the vector $\tilde{\psi}_{k-1}(\mathbf{z})$ is an $\ell_2$-normalized version of $\psi_{k-1}(\mathbf{z})$. This operation can be interpreted as a spatial convolution of the map $\xi_{k-1}$ with the filters $\mathbf{w}_{kl}$ followed by pointwise non-linearities;

2: set $\beta_k$ to be $\gamma_k$ times the spacing between two pixels in $\Omega_{k-1}$;

3: **feature pooling:** $\Omega_k'$ is obtained by subsampling $\Omega_{k-1}$ by a factor $\gamma_k$ and we define a new map $\xi_k : \Omega_k' \to \mathbb{R}^{p_k}$ obtained from $\zeta_k$ by linear pooling with Gaussian weights:

$$\xi_k : \mathbf{z} \mapsto \sqrt{2/\pi} \sum_{\mathbf{u} \in \Omega_{k-1}} e^{-\frac{1}{\beta_k^2}\|\mathbf{u}-\mathbf{z}\|_2^2} \zeta_k(\mathbf{u}). \tag{6}$$

**output** $\xi_k : \Omega_k' \to \mathbb{R}^{p_k}$ (new map);

---

**Approximation principles.** We proceed recursively to show that the kernel approximation property **(A)** is satisfied; we assume that **(B)** holds at layer $k$–1, and then, we show that **(A)** and **(B)** also hold at layer $k$. This is sufficient for our purpose since we have previously assumed **(B)** for the zeroth layer. Given two images feature maps $\varphi_{k-1}$ and $\varphi_{k-1}'$, we start by approximating $K(\varphi_{k-1}, \varphi_{k-1}')$ by replacing $\varphi_{k-1}(\mathbf{z})$ and $\varphi_{k-1}'(\mathbf{z}')$ by their finite-dimensional approximations provided by **(B)**:

$$K(\varphi_{k-1}, \varphi_{k-1}') \approx \sum_{\mathbf{z},\mathbf{z}' \in \Omega_{k-1}} \|\psi_{k-1}(\mathbf{z})\|_2 \|\psi_{k-1}'(\mathbf{z}')\|_2 e^{-\frac{1}{2\beta_k^2}\|\mathbf{z}-\mathbf{z}'\|_2^2} e^{-\frac{1}{2\sigma_k^2}\|\tilde{\psi}_{k-1}(\mathbf{z})-\tilde{\psi}_{k-1}'(\mathbf{z}')\|_2^2}. \tag{7}$$

Then, we use the finite-dimensional approximation of the Gaussian kernel involving $\sigma_k$ and

$$K(\varphi_{k-1}, \varphi_{k-1}') \approx \sum_{\mathbf{z},\mathbf{z}' \in \Omega_{k-1}} \zeta_k(\mathbf{z})^\top \zeta_k'(\mathbf{z}') e^{-\frac{1}{2\beta_k^2}\|\mathbf{z}-\mathbf{z}'\|_2^2}, \tag{8}$$

where $\zeta_k$ is defined in (5) and $\zeta_k'$ is defined similarly by replacing $\tilde{\psi}$ by $\tilde{\psi}'$. Finally, we approximate the remaining Gaussian kernel by uniform sampling on $\Omega_k'$, following Section 3.1. After exchanging sums and grouping appropriate terms together, we obtain the new approximation

$$K(\varphi_{k-1}, \varphi_{k-1}') \approx \frac{2}{\pi} \sum_{\mathbf{u} \in \Omega_k'} \left( \sum_{\mathbf{z} \in \Omega_{k-1}} e^{-\frac{1}{\beta_k^2}\|\mathbf{z}-\mathbf{u}\|_2^2} \zeta_k(\mathbf{z}) \right)^\top \left( \sum_{\mathbf{z}' \in \Omega_{k-1}} e^{-\frac{1}{\beta_k^2}\|\mathbf{z}'-\mathbf{u}\|_2^2} \zeta_k'(\mathbf{z}') \right), \tag{9}$$

where the constant $2/\pi$ comes from the multiplication of the constant $2/(\pi\beta_k^2)$ from (3) and the weight $\beta_k^2$ of uniform sampling orresponding to the square of the distance between two pixels of $\Omega_k'$.[4] As a result, the right-hand side is exactly $\langle \xi_k, \xi_k' \rangle$, where $\xi_k$ is defined in (6), giving us property **(A)**. It remains to show that property **(B)** also holds, specifically that the quantity (2) can be approximated by the Euclidean inner-product $\langle \psi_k(\mathbf{z}_k), \psi_k'(\mathbf{z}_k') \rangle$ with the patches $\psi_k(\mathbf{z}_k)$ and $\psi_k'(\mathbf{z}_k')$ of shape $\mathcal{P}_k'$; we assume for that purpose that $\mathcal{P}_k'$ is a subsampled version of the patch shape $\mathcal{P}_k$ by a factor $\gamma_k$.

We remark that the kernel (2) is the same as (1) applied to layer $k$–1 by replacing $\Omega_{k-1}$ by $\{\mathbf{z}_k\}+\mathcal{P}_k$. By doing the same substitution in (9), we immediately obtain an approximation of (2). Then, all Gaussian terms are negligible for all $\mathbf{u}$ and $\mathbf{z}$ that are far from each other—say when $\|\mathbf{u}-\mathbf{z}\|_2 \geq 2\beta_k$. Thus, we may replace the sums $\sum_{\mathbf{u}\in\Omega'_k}\sum_{\mathbf{z},\mathbf{z}'\in\{\mathbf{z}_k\}+\mathcal{P}_k}$ by $\sum_{\mathbf{u}\in\{\mathbf{z}_k\}+\mathcal{P}'_k}\sum_{\mathbf{z},\mathbf{z}'\in\Omega_{k-1}}$, which has the same set of "non-negligible" terms. This yields exactly the approximation $\langle\psi_k(\mathbf{z}_k),\psi'_k(\mathbf{z}'_k)\rangle$.

**Optimization.** Regarding problem (4), stochastic gradient descent (SGD) may be used since a potentially infinite amount of training data is available. However, we have preferred to use L-BFGS-B [9] on $300\,000$ pairs of randomly selected training data points, and initialize $\mathbf{W}$ with the K-means algorithm. L-BFGS-B is a parameter-free state-of-the-art batch method, which is not as fast as SGD but much easier to use. We always run the L-BFGS-B algorithm for $4\,000$ iterations, which seems to ensure convergence to a stationary point. Our goal is to demonstrate the preliminary performance of a new type of convolutional network, and we leave as future work any speed improvement.

## 4 Experiments

We now present experiments that were performed using Matlab and an L-BFGS-B solver [9] interfaced by Stephen Becker. Each image is represented by the last map $\xi^k$ of the CKN, which is used in a linear SVM implemented in the software package LibLinear [16]. These representations are centered, rescaled to have unit $\ell_2$-norm on average, and the regularization parameter of the SVM is always selected on a validation set or by 5-fold cross-validation in the range $2^i$, $i = -15\ldots,15$.

The patches $\mathcal{P}'_k$ are typically small; we tried the sizes $m \times m$ with $m = 3,4,5$ for the first layer, and $m = 2,3$ for the upper ones. The number of filters $p_k$ in our experiments is in the set $\{50,100,200,400,800\}$. The downsampling factor $\gamma_k$ is always chosen to be 2 between two consecutive layers, whereas the last layer is downsampled to produce final maps $\xi_k$ of a small size—say, $5\times5$ or $4\times4$. For the gradient map $\varphi_0$, we approximate the Gaussian kernel $e^{(1/\sigma_1^2)\|\varphi_0(\mathbf{z})-\varphi'_0(\mathbf{z}')\|_{\mathcal{H}_0}}$ by uniformly sampling $p_1 = 12$ orientations, setting $\sigma_1 = 2\pi/p_1$. Finally, we also use a small offset $\varepsilon$ to prevent numerical instabilities in the normalization steps $\tilde{\psi}(\mathbf{z}) = \psi(\mathbf{z})/\max(\|\psi(\mathbf{z})\|_2,\varepsilon)$.

### 4.1 Discovering the Structure of Natural Image Patches

Unsupervised learning was first used for discovering the underlying structure of natural image patches by Olshausen and Field [23]. Without making any a priori assumption about the data except a parsimony principle, the method is able to produce small prototypes that resemble Gabor wavelets—that is, spatially localized oriented basis functions. The results were found impressive by the scientific community and their work received substantial attention. It is also known that such results can also be achieved with CNNs [25]. We show in this section that this is also the case for convolutional kernel networks, even though they are not explicitly trained to reconstruct data.

Following [23], we randomly select a database of $300\,000$ whitened natural image patches of size $12 \times 12$ and learn $p = 256$ filters $\mathbf{W}$ using the formulation (4). We initialize $\mathbf{W}$ with Gaussian random noise without performing the K-means step, in order to ensure that the output we obtain is not an artifact of the initialization. In Figure 3, we display the filters associated to the top-128 largest weights $\eta_l$. Among the 256 filters, 197 exhibit interpretable Gabor-like structures and the rest was less interpretable. To the best of our knowledge, this is the first time that the explicit kernel map of the Gaussian kernel for whitened natural image patches is shown to be related to Gabor wavelets.

### 4.2 Digit Classification on MNIST

The MNIST dataset [22] consists of $60\,000$ images of handwritten digits for training and $10\,000$ for testing. We use two types of initial maps in our networks: the "patch map", denoted by CNK-PM and the "gradient map", denoted by CNK-GM. We follow the evaluation methodology of [25]

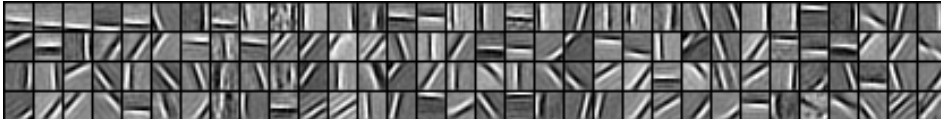

Figure 3: Filters obtained by the first layer of the convolutional kernel network on natural images.

| Tr. size | CNN [25] | Scat-1 [8] | Scat-2 [8] | CKN-GM1 (12/50) | CKN-GM2 (12/400) | CKN-PM1 (200) | CKN-PM2 (50/200) | [32] | [18] | [19] |
|---|---|---|---|---|---|---|---|---|---|---|
| 300 | 7.18 | 4.7 | 5.6 | 4.39 | 4.24 | 5.98 | **4.15** | | NA | |
| 1K | 3.21 | 2.3 | 2.6 | 2.60 | **2.05** | 3.23 | 2.76 | | NA | |
| 2K | 2.53 | **1.3** | 1.8 | 1.85 | 1.51 | 1.97 | 2.28 | | NA | |
| 5K | 1.52 | **1.03** | 1.4 | 1.41 | 1.21 | 1.41 | 1.56 | | NA | |
| 10K | 0.85 | **0.88** | 1 | 1.17 | **0.88** | 1.18 | 1.10 | | NA | |
| 20K | 0.76 | 0.79 | **0.58** | 0.89 | 0.60 | 0.83 | 0.77 | | NA | |
| 40K | 0.65 | 0.74 | 0.53 | 0.68 | **0.51** | 0.64 | 0.58 | | NA | |
| 60K | 0.53 | 0.70 | 0.4 | 0.58 | **0.39** | 0.63 | 0.53 | 0.47 | 0.45 | 0.53 |

Table 1: Test error in $\%$ for various approaches on the MNIST dataset without data augmentation. The numbers in parentheses represent the size $p_1$ and $p_2$ of the feature maps at each layer.

for comparison when varying the training set size. We select the regularization parameter of the SVM by 5-fold cross validation when the training size is smaller than $20\,000$, or otherwise, we keep $10\,0000$ examples from the training set for validation. We report in Table 1 the results obtained for four simple architectures. CKN-GM1 is the simplest one: its second layer uses $3 \times 3$ patches and only $p_2 = 50$ filters, resulting in a network with $5\,400$ parameters. Yet, it achieves an outstanding performance of $0.58\%$ error on the full dataset. The best performing, CKN-GM2, is similar to CKN-GM1 but uses $p_2 = 400$ filters. When working with raw patches, two layers (CKN-PM2) gives better results than one layer. More details about the network architectures are provided in the supplementary material. In general, our method achieves a state-of-the-art accuracy for this task since lower error rates have only been reported by using data augmentation [11].

## 4.3 Visual Recognition on CIFAR-10 and STL-10

We now move to the more challenging datasets CIFAR-10 [20] and STL-10 [13]. We select the best architectures on a validation set of $10\,000$ examples from the training set for CIFAR-10, and by 5-fold cross-validation on STL-10. We report in Table 2 results for CKN-GM, defined in the previous section, without exploiting color information, and CKN-PM when working on raw RGB patches whose mean color is subtracted. The best selected models have always two layers, with $800$ filters for the top layer. Since CKN-PM and CKN-GM exploit a different information, we also report a combination of such two models, CKN-CO, by concatenating normalized image representations together. The standard deviations for STL-10 was always below $0.7\%$. Our approach appears to be competitive with the state of the art, especially on STL-10 where only one method does better than ours, despite the fact that our models only use 2 layers and require learning few parameters. Note that better results than those reported in Table 2 have been obtained in the literature by using either data augmentation (around $90\%$ on CIFAR-10 for [18, 30]), or external data (around $70\%$ on STL-10 for [28]). We are planning to investigate similar data manipulations in the future.

| Method | [12] | [27] | [18] | [13] | [4] | [17] | [32] | CKN-GM | CKN-PM | CKN-CO |
|---|---|---|---|---|---|---|---|---|---|---|
| CIFAR-10 | 82.0 | 82.2 | **88.32** | 79.6 | NA | 83.96 | 84.87 | 74.84 | 78.30 | 82.18 |
| STL-10 | 60.1 | 58.7 | NA | 51.5 | **64.5** | 62.3 | NA | 60.04 | 60.25 | 62.32 |

Table 2: Classification accuracy in $\%$ on CIFAR-10 and STL-10 without data augmentation.

## 5 Conclusion

In this paper, we have proposed a new methodology for combining kernels and convolutional neural networks. We show that mixing the ideas of these two concepts is fruitful, since we achieve near state-of-the-art performance on several datasets such as MNIST, CIFAR-10, and STL10, with simple architectures and no data augmentation. Some challenges regarding our work are left open for the future. The first one is the use of supervision to better approximate the kernel for the prediction task. The second consists in leveraging the kernel interpretation of our convolutional neural networks to better understand the theoretical properties of the feature spaces that these networks produce.

**Acknowledgments**

This work was partially supported by grants from ANR (project MACARON ANR-14-CE23-0003-01), MSR-Inria joint centre, European Research Council (project ALLEGRO), CNRS-Mastodons program (project GARGANTUA), and the LabEx PERSYVAL-Lab (ANR-11-LABX-0025).

## Footnotes

*LEAR team, Inria Grenoble, Laboratoire Jean Kuntzmann, CNRS, Univ. Grenoble Alpes, France.

[1]In the kernel literature, "feature map" denotes the mapping between data points and their representation in a reproducing kernel Hilbert space (RKHS) [26]. Here, feature maps refer to spatial maps representing local image characteristics at everly location, as usual in the neural network literature [22].

[2]When $\Omega$ is not discrete, the notation $\sum$ in (1) should be replaced by the Lebesgue integral $\int$ in the paper.

[3]For two sets $A$ and $B$, the Minkowski sum $A + B$ is defined as $\{a + b : a \in A, b \in B\}$.

[4] The choice of $\beta_k$ in Algorithm 2 is driven by signal processing principles. The feature pooling step can indeed be interpreted as a downsampling operation that reduces the resolution of the map from $\Omega_{k-1}$ to $\Omega_k$ by using a Gaussian anti-aliasing filter, whose role is to reduce frequencies above the Nyquist limit.

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
