[Reviews · NeurIPS 2014]

Submitted by Assigned_Reviewer_8

The authors propose a novel method for image representation called Convolutional Kernel Methods. It is different from similar methods in that it is not designed for explicitly reconstructing then data, or for classifying it. The patch-map and gradient-map approaches obtain quite competitive numbers on MNIST, and reasonable numbers (if not quite state of the art) on CIFAR-10 and STL-10. The Gabor filters obtained on the natural image patches are quite interesting too.

Quality: the paper has a novel approach that is well-described and the experiments are well-designed.

Clarity: I found the exposition quite dense, especially the parts regarding the linear approximation.

Originality: I think the paper presents a number of original ideas, I am not aware of very similar approaches in the literature.

Significance: I think the proposed method is sufficiently different and practical enough that others in the community could be interested in it.

One comment about the paper, which would make it more readable and understandable: it would be great if the authors could make a pseudo-code like algorithm that details the steps done for learning. This would greatly improve the readability of Sections 3.3 and 3.4.
Summary: A novel approach to building a kernel-based feature representation that is similar to what a convolutional neural network does, except the features are trained in an unsupervised way without an explicit reconstruction criterion. Overall, an interesting and solid piece of work, that should be of interest to the community despite the not-so-stella results on CIFAR and STL sets.

Submitted by Assigned_Reviewer_17

This paper presents a kernel based approach to learning a model that is heavily based on convolutional networks. The standard types of layers of a convolutional network is replicated by a kernel that produces an equivalent or similar response to that of original. I think the main aspect of the paper is the approximation of the gaussian kernel by a learned formulation given in equation 4, however the exact procedure for training is not clear from the paper. The results presented on widely used datasets suggest that despite unsupervised training of the hierarchy of feature extractor, the system achieves competitive results.

The method presented in the paper is algorithmically quite heavy compared to the simplicity of convolutional networks. On the other hand, it might provide useful insights into new models or better understanding of current models. I think the main drawback of this paper is that, it does not contain a clean explanation of an end-to-end process of training the model. Convolutional nets are popular since they achieve state of the art results on challenging datasets, but at the same time, the training procedure is very simple and scales to large datasets. I think it would be very helpful if the paper included a more clean explanation of the overall algorithm and computational requirements of training a CKN.
Summary: I think this is an interesting direction of work that could potentially be helpful to both communities. I think in its current form the paper is at times very confusing to read and missing important information (or clarifications) on the general properties of the end algorithm.

Submitted by Assigned_Reviewer_42

The paper proposes a novel criterion to train convolutional neural nets -- to approximate a kernel of pre-defined feature maps. The kernel is a convolutional gaussian kernel, i.e. one feature map is convolved with another feature map and the kernel value is the energy of this operation. Such a kernel is invariant to translation.

The contribution of the work consists of approximating the above kernel using a CNN. The kernel is approximated as a sequence of filtering and a non-linearity which is a layer in a CNN. The parameters of the filters are learned such that the kernel is approximated.

The authors provide a rigorous technical exposition followed by proof of concept empirical results on MNIST and CIFAR. In particular they seem to perform comparably to scattering NNs.

A downside of the work is that one has to define a kernel to approximate. The definition of the kernel involves definition of features, the core problem which NNs are good at solving in a data driven way. For example, in the conducted experiment the features are either gradients or normalized pixel values.
Summary: The paper proposes a novel approach on an important aspect of CNNs -- learning invariance. It is well explained and motivated and thus deserves presentation to a larger audience.
Author Feedback
Author rebuttal: We thank the reviewers for their insightful comments. We appreciate their positive remarks and their constructive criticisms, and we respond below to the different concerns they raised.

****************************
Reviewer_8:
We agree that the current manuscript is a bit dense, and we thank the reviewer for his suggestion to add a pseudo-code that will help the reader go through Sections 3.3 and 3.4. This will also illustrate the fact that our training procedure is very simple. Therefore, we will follow the reviewer's advice, and, more generally, we will make our best to make the paper more appealing and accessible to a larger audience, if it is accepted.

****************************
Reviewer_17:
- Regarding the clarity of the paper, we believe that our response to Reviewer_8 will also address Reviewer_17's comment.
- The reviewer finds our approach algorithmically heavy compared to classical CNNs. We disagree with this statement but we apologize if the submitted version of our paper is not clear enough regarding the computational aspect. We agree that establishing the link between the original kernel formulation and our convolutional kernel network (CKN) is non-trivial and requires slightly heavy notation. However, we argue that training our convolutional kernel networks is ***as simple as training CNNs***. More precisely, our training and testing pipelines are almost identical to CNNs, up to a different objective function. We have detailed below the training procedure and listed the differences with a CNN trained in an unsupervised manner: all the steps are the same, but the cost function is different. The computational complexity for computing our gradient is of the same order of magnitude of the ones used for unsupervised networks, e.g., [Ranzato et al.]. If the paper is accepted, we will also release our Matlab implementation at the time of publication. We (subjectively) find it short and easy to use, which should confirm the simplicity of our approach.

*** Training procedure for layer k***
Input: feature maps of layer k-1 for training images;
Output: feature maps of layer k for training images and layer parameters;
Step 1, patch-extraction: sample N training pairs of patches from the input feature maps;
Step 2, contrast-normalization: normalize the training patches and record their norm;
Step 3, layer parameters learning: minimize the cost function (4) evaluated on the N training pairs using L-BFGS (SGD would be probably faster but subject to high variability due to the choice of the step size, see [Ngiam et al.]);
Step 4: convolution with the learned filters of the input feature maps + pointwise non-linearity;
Step 5, pooling: Gaussian filtering and downsampling, producing the output feature maps.

*** Differences with unsupervised learning of CNNs ***
1) use of N pairs of training patches instead of N training patches;
2) use of a different cost function (4), sampled at N pairs.

************************
- Reviewer_42
We thank the reviewer for his positive comments. We respectfully disagree with the fact that needing to define input features for the first layer should be seen as a downside compared to traditional CNNs. We agree that using gradient features is not classical in CNNs, but using normalized pixel values simply corresponds to using contrast normalized images, which is a common practice for CNNs (see, e.g., [Wan et al.], where even more sophisticated pre-processing steps such as whitening are used on some datasets).

[Wan et al.] Regularization of neural networks using DropConnect. ICML. 2013.
[Ngiam et al.] On optimization methods for deep learning. ICML. 2011.
[Ranzato et al.] Unsupervised learning of invariant feature hierarchies with applications to object recognition. CVPR. 2007.